# Proximal Perineural Femoral Nerve Injection in Pigs Using an Ultrasound-Guided Lateral Subiliac Approach—A Cadaveric Study

**DOI:** 10.3390/ani11061759

**Published:** 2021-06-12

**Authors:** Robert Trujanovic, Pablo E. Otero, Peter Marhofer, Ulrike Auer, Silvio Kau

**Affiliations:** 1Anesthesia and Perioperative Intensive Care Unit, Department of Small Animals and Horses, University of Veterinary Medicine, 2210 Vienna, Austria; ulrike.auer@vetmeduni.ac.at; 2Department of Anesthesiology, Faculty of Veterinary Medicine, University of Buenos Aires, Buenos Aires 1421BA, Argentina; potero@fvet.uba.ar; 3Department of Anesthesia and Intensive Care Medicine, Orthopedic Hospital Speising, 2210 Vienna, Austria; peter.marhofer@live.com; 4Department of Anaesthesia, General Intensive Care Medicine and Pain Therapy, Medical University of Vienna, 2210 Vienna, Austria; 5Institute of Morphology, Working Group Anatomy, Department of Pathobiology, University of Veterinary Medicine Vienna, 2210 Vienna, Austria; silvio.kau@vetmeduni.ac.at

**Keywords:** swine, animal model, regional anesthesia, peripheral nerve block, femoral nerve block, lumbar plexus, orthopedic surgery

## Abstract

**Simple Summary:**

Desensitizing the femoral nerve improves pain control in several species undergoing pelvic limb surgeries. Despite possible advantages, this method has not yet been described in pigs, although they make an accepted surgical animal model. We developed an approach for femoral nerve blockade using ultrasound guidance in pigs which could be useful for pain control in pigs undergoing pelvic limb surgery.

**Abstract:**

Desensitizing the femoral nerve (FN) improves pain control in several species undergoing pelvic limb surgeries. Despite its advantages, this method has not yet been described in pigs, although they make an accepted surgical animal model. Based on anatomical dissections, first performed in two pig cadavers, an ultrasound-guided access for localization and perineural infiltration of the FN trunk at the iliopsoas compartment level was specified. The FN was found running between the psoas major and medial portion of iliac muscle. Ultrasonographically, the FN appeared as a hypoechogenic round–oval structure surrounded by a hyperechogenic rim. Technical feasibility and accuracy were assessed in six additional pig cadavers by injecting 0.15 mL kg^−1^ methylene blue dye bilaterally in direct proximity to the nerve. The needle was inserted caudoventral to the coxal tuber, traversing the ultrasound plane as it progressed towards the FN in dorsomedial direction. Staining of the nerve was evaluated by dissection. The injection was considered effective if the nerve was stained in its entire circumference over a length of 2 cm. FNs of all investigated individuals could be successfully stained. This ultrasound-guided subiliac approach allows feasible and accurate access to the FN and may be useful in producing a successful blockade in vivo.

## 1. Introduction

Pigs are a widely accepted translational animal model that is used in a growing number of experimental orthopedic surgical interventions for humans [1,2,3]. Adequate pain management, however, has not received enough attention to date. 

Surgical interventions lead to a complex stress-response, characterized by endocrine-metabolic alterations including elevated resting energy expenditure, as well as an inflammatory response that may cause immunosuppression [4,5]. Surgical stress-response puts the patient at high risk for delayed wound healing and, depending on the severity of applied damage, even organ failure and death [6].

The incorporation of regional anesthesia in balanced anesthetic protocols has been shown to promote better perioperative outcomes and reduced opioid consumption, morbidity, mortality, and hospitalization time in humans [7,8]. Similarly, animals benefit from regional anesthesia [5,9,10]. In dogs undergoing pelvic limb orthopedic procedures, regional anesthesia using femoral nerve (FN) and sciatic nerve (ScN) PNBs has been associated with improved quality of recovery and low perioperative levels of stress biomarkers, i.e., cortisol and glycemia. The combination of general anesthesia and opioids lead to bad recovery quality and up to threefold increase of stress biomarkers [5].

Peripheral nerve blocks are a type of regional anesthesia in which the local anesthetic is injected near a specific nerve or bundle of nerves to block sensations of pain from a specific area of the body. They are used for pre-emptive and multimodal pain control, e.g., to alleviate peri- and postoperative pain in both humans and animals [11,12]. These techniques may provide cost-effective analgesia and minimize considerable adverse effects even in pigs [13]. However, compared to small animals [14,15,16], studies on PNBs in pigs are scarce. 

The introduction of ultrasound (US) technology to perform PNBs is known to have increased their success rate. Ultrasound-guidance helps to accelerate the block onset, increase the duration of action, and decrease the quantity of local anesthetic (LA) used to minimize the risk for LA-induced toxicity [17,18,19,20,21].

A recent study in pigs described the ultrasonographic appearance of the ScN and performed a US-guided approach to block this nerve [22]. However, to our knowledge, there are no similar studies regarding the FN in pigs. A detailed description of regional anatomical structures and a valid US-guided technique to block the FN in pigs could be useful for effective regional anesthesia in orthopedic surgical procedures of pelvic limbs. 

In this study, we attempted to describe a US-guided lateral subiliac approach to block the FN in pigs. The first study phase compiled of a descriptive gross anatomical dissection of the FN and its adjacent anatomical structures, e.g., nerves, vessels, musculoskeletal and fascial elements; an optimal injection side and angle were assessed. In the second phase, relevant anatomical features were located using US, and an in-plane injection was performed under US guidance and visualization of methylene blue dye spread in near proximity to the target nerve upon dissection. 

## 2. Materials and Methods

### 2.1. Animals and Design

The study design was divided into two phases. In the first phase, fresh cadavers of *n* = 2 male F1 cross breed German Large White and German Landrace pigs (12 weeks of age; 37 and 39 kg) were used to conduct explorative ultrasonographic scans and a gross anatomical dissection. In the second phase, fresh cadavers of *n* = 6 male F1 cross breed German Large White and German Landrace pigs (12 weeks of age; median body weight of 36 kg (range 29–39) were used to conduct a US-guided lateral subiliac injection of methylene blue dye in close proximity to the FN. All pigs were obtained from a non-related non-survival study which did not involve lumbosacral or pelvic limb procedures. Pigs were euthanized humanely by intravenous injection of T61 (Merck Sharp and Dohme, Kenilworth, NJ, USA) under general anesthesia. 

### 2.2. Sonoanatomical Study and Anatomical Dissection

Dissection of the left and right FN and adjacent anatomical structures was performed using either of the two pig cadavers intended for preliminary anatomical assessment (fresh cadaver; six lumbar vertebrae). Before dissection, the sonoanatomy of the hypaxial lumbosacral region was determined on transversal ventrodorsal directed ultrasonographic-scans. Then a ventral midline incision of the skin, linea alba, and peritoneum was made. The abdominal organs were removed for better visualization of hypaxial structures in the retroperitoneal space. Following landmarks served to identify the FN, adapt US beam orientation, and choose needle insertion direction: coxal tuber, corpora of the last two lumbar vertebrae, terminal branches of the abdominal aorta and caudal vena cava, psoas major and minor muscles, medial and lateral portion of the iliacus muscle, sartorius muscle, iliac fascia, and muscular lacuna. Tracking back the FN to the intervertebral foramina of the supplying lumbar nerves (lumbar plexus) served to identify the segmental origin and course of the nerve. A second cadaver (freshly frozen; six lumbar vertebrae) was used for transversal of serial frozen sections similar to the ascertained optimum US beam direction to correlate native anatomical structures with acquired US images.

### 2.3. US-Guided Injection Procedure

Before injections, ultrasonographic scans of the left and right FN, and adjacent anatomical structures were performed in all six pig cadavers intended for the second study phase. The scans were carried out immediately after euthanasia. The cadavers were placed in a slightly slanted dorsal recumbency with the side of interest facing more upwards. The hair from the lumbar area was clipped, the skin cleaned, and US coupling gel (Softa-Man, ViscoRub, B. Braun, Austria) applied. A portable US machine (NanoMaxx, Fujifilm SonoSite, Bothell, WA, USA) and a 38-mm broadband linear array US transducer (L38n/10–5 MHz Transducer, Fujifilm SonoSite, Bothell, WA, USA) were used to identify the anatomical reference structures and target nerve. A scan depth setting of 4.9 cm was selected to perform the imaging and further injection guidance. The transducer was positioned perpendicular to the long axis of the spine, ventral, and slightly medial to the respective coxal tuber (Figure 1). The coxal tubera was palpated manually. To locate and trace the projection of the FN trunk, the transducer was slid cranially and caudally from this point. The FN trunk was specified as the most proximal portion of the nerve, immediately after convergence of main supplying lumbar plexus nerves. Once all sonoanatomical landmarks and the FN trunk were identified, an atraumatic PNB needle (Stimuplex, B-Braun, Maria Enzersdorf, Austria) was inserted from lateral, entering 1 cm caudal and ventral to the coxal tuber of os ilium, and advanced under US guidance (in-plane) in close proximity to the FN. The needle insertion position and needle inclination are displayed in Figure 1. Subsequently, a 1 mL test volume of an aqueous methylene blue dye (1% *w*/*v*; methylene blue, Alfa Aesar, Thermo Fisher, Dreieich, Germany) was injected to test for correct needle positioning. If the spread of the dye was appropriate, the remaining volume was injected (total 0.15 mL kg^−1^).

The cadavers were dissected 10 min after the injection, and the FNs evaluated. The dissection followed the same criteria as mentioned for the anatomical study. The presence of methylene blue dye around the target nerve for a length of ≥2 cm was considered a successful injection [23].

## 3. Results

### 3.1. Anatomical Disection Helped Define a Suitable US Window for Proximal FN Injections

In order to assess an adequate acoustic window for US-guided injection of the proximal FN in the lumbosacral region, an anatomical dissection of the nerve and adjacent hypaxial structures was performed. In the area of interest, both the left and right FN could be identified in all pig cadavers used for anatomical dissection (*n* = 2). In both pigs, the ventral branch of the fifth lumbar segmental nerve represented the main supply to the FN. However, plexus forming ventral branches of the fourth (L4) and sixth (L6) lumbar nerves were also involved. With regard to their intervertebral origin, the segmental nerves were numbered according to the number of the lumbar vertebra cranial to the nerve. 

The proximal FN trunk was observed at the level of the body of the sixth lumbar vertebra, running between the psoas major and medial portion of the iliacus muscle. The latter also had a lateral portion. Both portions of the iliac muscle encompass the psoas major muscle, together representing the iliopsoas muscle. We assessed the connective tissue area between the medial iliacus muscle portion and psoas major muscle as the iliopsoas compartment. All relevant lumbar plexus nerves supplying the FN (L4–L6) were found running through there. The FN ran in a caudoventral direction through the iliopsoas compartment to emerge from the compartment covered by the iliac fascia ventrally at the level of the lumbosacral junction. The nerve gave off large muscle branches before exiting the abdomen through the muscular lacuna. All relevant anatomical structures are displayed in Figure 2. 

Based on the anatomical conditions, the area ventral and 1-cm caudal to the most ventral aspect of the respective coxal tuber was chosen as the appropriate injection site (Figure 2b–d). Here the FN was found running deep in the iliopsoas compartment. It was believed that injecting there could reduce the risk of accidental intra-abdominal injection or intra-vasal injection (external iliac artery, common iliac vein) through superficial access. It was further assumed that by injecting there, the risk of injury to the nerves and vessels that run ventral on the iliopsoas muscle to the lateral side a few centimetres further caudal, i.e., genitofemoral nerve (L3), cutaneous femoris lateralis nerve (L4), deep circumflex iliac artery and vein, could be decreased (Figure 2c).

### 3.2. Anatomical Landmarks Can Be Depicted in Detail by Ultrasonography 

For targeted perineural FN injections in the iliopsoas compartment, the FN and adjacent anatomical structures first had to be correctly displayed using US. Cranioventral aspects of the iliac crest (most ventral aspects of the coxal tuber) could be easily palpated and considered as useful landmarks for correct transducer positioning in all cases. The transducer was oriented perpendicular to the spine at the level 1 cm caudal to the coxal tuber and facing dorsally (Figure 1), which resulted in an appropriate US window in all cadavers. The iliopsoas muscle was observed as an ovoid to triangular structure with an internal pattern of scattering echoes in the transverse scans. All relevant muscles, i.e., psoas major and minor muscle, medial portion of the iliac muscle, and quadratus lumborum muscle, could be identified. The iliac artery and vein were identified as circular hypoechoic structures located medial to the interface of the iliopsoas and psoas minor muscles. The vertebral body and respective transverse process of the sixth lumbar vertebra were seen as a hyperechoic line producing acoustic shadowing. 

The FN was correctly identified by US in the suggested area. The nerve appeared as a single, well-distinguishable, round, hypoechogenic structure surrounded by a marked hyperechogenic rim in the transverse scans (Figure 3). The FN could be located within the iliopsoas compartment, i.e., between the psoas major and medial portion of the iliac muscle. By changing the position of the transducer further cranially or caudally, the FN appeared either further dorsally or ventrally within the iliopsoas compartment. This can be explained by its craniodorsal–caudoventral course, which could also be shown in the dissected specimens. Overall, there was a good correlation between the US images and the cryosections (Figure 2b).

### 3.3. Perineural FN Injections

Based on the anatomical observations, methylene blue dye was injected in the iliopsoas compartment in near proximity to the FN under US guidance. The staining pattern was evaluated after gross dissection. The dye was observed staining successfully both FNs in all six cases (Figure 4). All FNs were stained over a distance of >2 cm in their entire circumference. Distribution of the injectate from the injection site was observed to occur simultaneously in cranial and caudal direction. There were no signs of intraperitoneal, retroperitoneal, or intravascular injectate spread.

## 4. Discussion

This study aimed to describe the gross anatomical and ultrasonographic appearance of the proximal FN in the direct hypaxial lumbosacral region in pigs. The information further served to develop a US-guided approach for injecting a defined volume of methylene blue dye into the iliopsoas compartment in direct proximity to the FN. Methodological feasibility and accuracy were assessed by means of bilateral injections in six different pigs. To our knowledge, this is the first study investigating the use of US for location and perineural injection of the FN in the iliopsoas compartment in pigs. 

The preceding anatomical study was useful to establish an adequate acoustic window for subsequent injections. The anatomical features of the FN in the iliopsoas compartment, and adjacent structures in the lumbosacral region were similar to previous descriptions in small mammals [11]. The ventral branches of the lumbar spinal nerves joined together in the iliopsoas compartment forming the lumbar plexus [24]. In the pigs used for anatomical dissection, we found that branches of L4–L6 supplied the FN. This result deviates slightly from the information in literature. According to a standard anatomical textbook, only L5 and L6 supply the FN in pigs [24]. Herrera et al. (2018) reported that L5 and L6 supply the FN in only 26.7% of investigated pig foetuses, whereas 66.7% showed L4 and L5, and 6.6% L3 and L4 to supply the FN [25]. We found that the branch provided by L4 was a really delicate nerve. It can be difficult to find such thin nerves in foetuses. Segmental involvement may also be related to the breed or number of lumbar vertebrae. The latter assumption is not supported by the fact that, like in all our pigs, 96.7% (n = 29) of those in the study of Herrera et al. (2018) had six lumbar vertebrae [25]. The number of lumbar segments involved in the FN, and thus the width of its area of origin, could result in insufficient blockade if the injection, for example, is too far cranial. This shows the importance of reaching the FN as targeted as possible and after union of involved lumbar nerves.

When the FN block was performed in dogs, using a US-guided technique, a 50% failure rate was reported using a femoral triangle approach [26]. Contrarily, the same authors reported a success rate of 100% when a suprainguinal approach to this nerve was employed [15]. In our study, using a similar approach, perineural injection was successful in 100% of cases. The low success rate in the femoral triangle approach has been related to difficulties in locating the FN by US owing to the great amount of connective tissue around the FN in this area, the reduced nerve length, and its branches at this level [26]. The suprainguinal approach locates the FN within the psoas/iliopsoas muscle at its largest diameter before ramification. This may improve visualization and therefore explain the higher reported success rate [26]. Similarly, the lateral subiliac approach employed in our pig cadaver study located the main trunk of the FN within the iliopsoas compartment. 

Both in small mammals and humans, the FN has been described as a hyperechogenic triangular structure when it is observed by US at the level of the femoral triangle [26,27]. In our study, the FN within the iliopsoas muscle was observed by US as a hypoechogenic oval–round structure surrounded by a hyperechogenic rim. These differences may be explained by the effect of the tissues covering the FN in different anatomical regions. The FN is covered by multiple fascial planes and fat tissue at the level of the femoral triangle. While it is imbedded in delicate intermuscular fat, connective tissue, and is completely surrounded by muscle at the level of the iliopsoas compartment [11,15]. The sonoanatomical landmarks defined to localize the FN trunk were found approximately 1 cm caudal to the projection of the coxal tuber, which could be easily palpated. In larger pigs, however, palpation could be more difficult due to the more developed musculature and thick subcutaneous fat. Here it might be necessary to look for other superficial landmarks, or to focus entirely on the US-based orientation. In humans, obesity can impair ultrasonographic landmark visualization leading to failure in regional anesthesia [28]. Although the localization of the proximal FN by the US-guided lateral subiliac approach was successful in all pigs used in our study (6 pigs; 12/12 cases), the procedural feasibility has not yet been assessed in larger pigs.

With this approach it was possible to observe the needle and the nerve in the same plane, which is a significant advantage. The visualization of the needle shaft has been considered necessary to accomplish a safer approach to a peripheral nerve [29]. The visualization of the target nerve in a gross-sectional view further allows a detailed observation of the distribution of the anesthetic solution around the nerve [30]. This can help reduce the volume of LA for PNBs, thereby minimizing the incidence of LA-related toxicity and possible nerve damage. We used a total volume of 0.15 mL kg^−1^ per injection side. This volume has been described and recommended for a similar approach in distinct small animal species [11]. Whether even smaller volumes are sufficient to successfully block only the FN in pigs, or whether larger amounts are needed, must be investigated by means of in vivo studies.

The length of the nerve in contact with the LA is a major factor in determining the success of a PNB. In myelinated nerves, at least three nodes of Ranvier must be exposed to local anesthetic to ensure that nerve conduction is blocked, this corresponds to 3–4 mm of nerve [23]. An in vitro study suggested that staining of ≥2 cm along a peripheral nerve should be considered sufficient to produce a clinically effective nerve blockade [23]. Baldo et al. (2018) reported that continuous staining over a length of >1 cm should already be sufficient [31]. The volume we used in combination with the US-guided injection was more than enough to successfully stain the FN in all cadavers. Accordingly, a volume reduction could be considered. 

This cadaveric study only allows us to speculate on the clinical efficacy of the conducted perineural FN injections. The results from this study showed, however, that the US-guided lateral subiliac approach to the FN in pigs produced an adequate methylene blue dye distribution, so that an adequate nerve block can be assumed when using a LA. Due to the lack of distinct fascia between the muscles of the iliopsoas compartment, and around the nerves located there, it is very likely that the injected LA will reach the perineural space, even if the tip of the needle cannot be advanced in the direct proximity to the nerve. For this reason, this block is usually very effective in small animals [11].

In this study, evidence of intraperitoneal, retroperitoneal, or intravascular injection of methylene blue dye was not seen in any case. A similar study in cats showed a small amount of LA in the retroperitoneal space in 3 out of four cases [16].

## 5. Conclusions

The described US-guided lateral subiliac dye injection approach resulted in successful staining of the proximal FN within the iliopsoas compartment in 100% of cases. This can be a feasible and accurate technique for in vivo antinociceptive blockade of the FN receptive field in pigs. 

## Figures and Tables

**Figure 1 animals-11-01759-f001:**
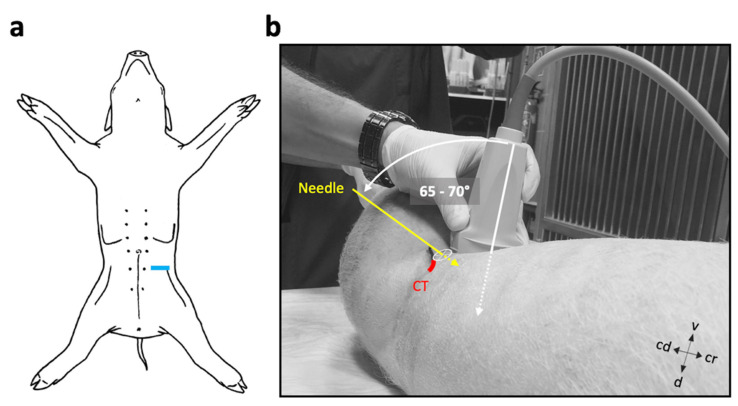
Ultrasound-transducer positioning and suggested insertion direction of the PNB-needle. (**a**) Transducer position in ventral view. (**b**) The US beam is directed dorsally and slightly medially; the orientation marker on the transducer faces lateral. The needle was inserted from lateral, entering 1 cm caudal and ventral to the coxal tuber (CT) of os ilium, using an in-plane technique and displayed angle. cr, cranial; cd, caudal; v, ventral; d, dorsal.

**Figure 2 animals-11-01759-f002:**
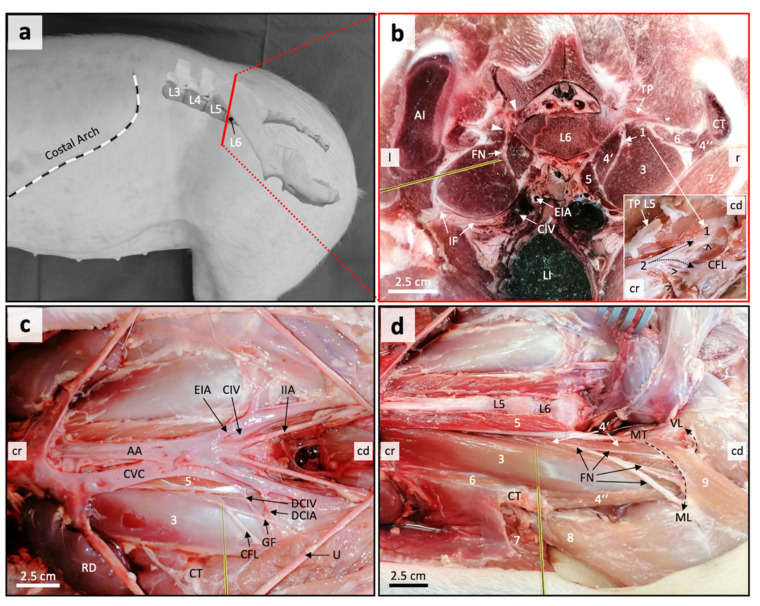
Gross anatomical dissection of the proximal femoral nerve (FN) and adjacent musculoskeletal and neurovascular structures. The proposed position and direction for needle insertion is shown in yellow. (**a**) Left lateral view showing the assumed section plane of the transverse frozen section in b; L3–L6, third to sixth lumbar vertebrae. (**b**) Slightly oblique transverse frozen section, caudal view. Left side (l) showing the ala of ilium (AI) and fifth lumbar nerve (arrowheads), which is the main segmental nerve supplying the FN. Right side (r) showing the coxal tuber (CT) and caudal directed plexus branch (1; insert: 1) of the fourth lumbar nerve (insert: 2) to supply the FN. The insert shows a ventrolateral view on nerve structures after removal of the psoas major muscle at the level of L5 to L6 using a fresh cadaver. TP L5, transverse process of L5; muscular nerve branches (open arrowheads); CFL, cutaneous femoris lateralis nerve; cr, cranial; cd, caudal. Further structures of the transverse section: EIA, external iliac artery; CIV, common iliac vein; IF, iliac fascia; LI, large intestine; 3 = psoas major muscle; 4 = iliac muscle, 4′ = its medial and 4″ = its lateral part; 5 = psoas minor muscle with its prominent insertion tendon; 6 = quadratus lumborum muscle; 7 = abdominal wall muscles (yellow line). (**c**) Ventral view of relevant hypaxial structures at the level of the iliopsoas compartment after removal of transversal and iliac fascia elements and fat using a fresh cadaver. AA, abdominal aorta; CVC, caudal vena cava; IIA, internal iliac artery; DCIA, deep circumflex iliac artery; DCIV, deep circumflex iliac vein; GF, genitofemoral nerve; RD, ren dexter; U, ureter. The numbering applies as in b. (**d**) Ventral view on the FN and its branching pattern after removal of vascular structures. MT, muscular tubercle of psoas minor and sartorius muscle; ML, muscular lacuna; VL, vascular lacuna; 8 = tensor fasciae latae muscle; 9 = sartorius muscle. The remaining numbering applies as in b.

**Figure 3 animals-11-01759-f003:**
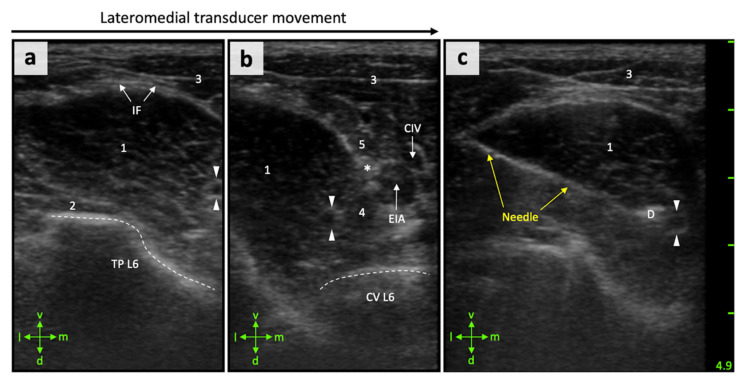
Transversal ultrasound images showing sonoanatomical reference structures and perineural injection of a right femoral nerve. (**a**) Lateral aspect, (**b**) medial aspect, and (**c**) with the inserted needle in plane. IF, iliac fascia; TP L6, transverse process of the sixth lumbar vertebra (bone surface = dashed line); CV L6, vertebral corpus of L6 (bone surface = dashed line); CIV, common iliac vein; EIA, external iliac artery; D, dye; femoral nerve (arrowheads); 1 = psoas major muscle; 2 = quadratus lumborum muscle; 3 = abdominal wall muscles; 4 = medial part of iliac muscle; 5 = psoas minor muscle and its tendon (asterisk); l, lateral; m, medial; v, ventral; d, dorsal. Depth marks (1 cm) apply to images a–c.

**Figure 4 animals-11-01759-f004:**
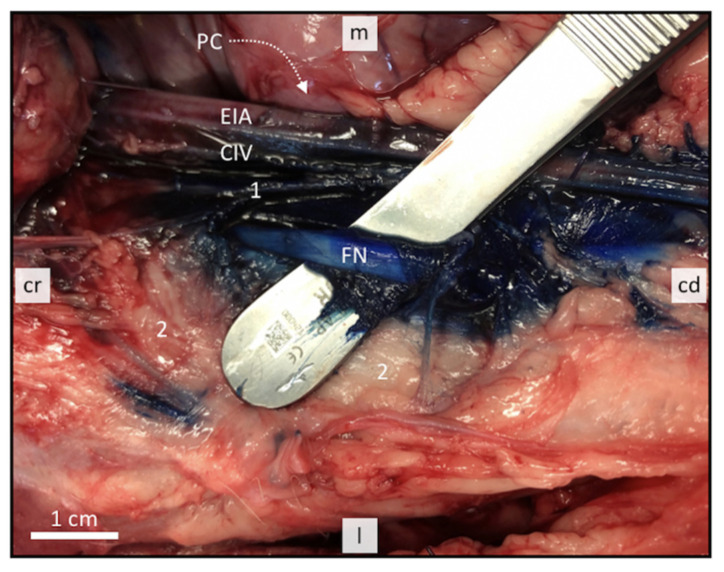
Dissection of a successfully stained right femoral nerve after lateral subiliac iliopsoas compartment injection with 1% methylene blue dye. The nerve was mobilized from the iliopsoas compartment to assess staining. 1 = psoas minor muscle; 2 = psoas major muscle and medial part of iliac muscle covered by fat; PC, pelvic cavum; EIA, external iliac artery; CIV, common iliac vein; FN, femoral nerve; cr, cranial; cd, caudal; m, medial; l, lateral.

## Data Availability

The data presented in this study are available on request from the corresponding author.

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
