# Peer review of "Proximal Perineural Femoral Nerve Injection in Pigs Using an Ultrasound-Guided Lateral Subiliac Approach—A Cadaveric Study"

_animals, 2021, doi:10.3390/ani11061759_

Round 1

Reviewer 1 Report

Dear Authors,

Thank you very much for this well-designed study. All the relevant information is included as the rationale of the study. I have only a few suggestions that can be addressed directly at the time of editorial editing.

Have you performed the US procedures on both sides for each swine? If so, probably it is worthing to mention. 
Line 271 You mentioned accuracy, but you do not define it before. 
Line 284 You report “… L4 was a really delicate nerve”; what do you mean? That it is easy to damage or it is tiny?
Line325 appr. ?
Line 335 may be “….shaft has been….”

Author Response

Dear reviewer,

Thank you for constructive comments concerning our manuscript entitled “Proximal perineural femoral nerve injection in pigs using an ultrasound-guided lateral subiliac approach - A cadaveric study”. We have studied your comments carefully and made a correction which we hope to meet with your approval. We answer your questions or comments in details in the following texts. Detailed answer to review:

Have you performed the US procedures on both sides for each swine? If so, probably it is worthing to mention. -Line 234-236: Before injections, ultrasonographic scans of the left and right FN and adjacent anatomical structures were performed in all six pig cadavers intended for the second study phase. Thank you! Line 271 You mentioned accuracy, but you do not define it before. - I am not sure that I understood the question/comment. Please send the entire sentence so that I can correct it. Thank you! Line 284 You report “… L4 was a really delicate nerve”; what do you mean? That it is easy to damage or it is tiny? - Thank you for this comment. Yes, that it is tiny. Should we rephrase it? Line325 appr. ?  -Thank you! Line 335 may be “….shaft has been….” -Thank you very much!  

We really hope manuscript modification can meet with your approval. Thank you very much.

Sincerely,

XXX

Reviewer 2 Report

Dear Author/s,

It has been a pleasure to read about your cadaveric study.

I am so grateful few clinicians are working in species different from canine and feline. Also, your article will help about postoperative pain management in swine research projects.

I only have one suggestion, and it is about your sentence in the first paragraph of the introduction: "Adequate pain management, however, has received litle attention to date". I do not 100% agree with that sentence due to perioperative management is important during research animals (check Bradbury et al. "pain management in pigs undergoing experimental surgery", 2016). However, it is true that both postoperative and locoregional techniques are less likely or even lacking in the protocols. I would rephrase that sentence.

I do not have any other suggestions. It is a great study.

Kind regards

Author Response

Dear reviewer,

Thank you for constructive comments concerning our manuscript entitled “Proximal perineural femoral nerve injection in pigs using an ultrasound-guided lateral subiliac approach - A cadaveric study”.

We rephrased the sentence: Line 84-84: Adequate pain management, however, has not received enough attention to date.  

We really hope manuscript modification can meet with your approval. Thank you very much.

Sincerely,

XXX

Reviewer 3 Report

This is an interesting study on the use of ultrasound to detect the location of the femoral nerve in pigs using cadavers.

The Paper is far to long and needs extensive editing.

The Introduction is rambling and should be  significantly reduced in length as should the Discussion and concentrate on the main points of the findings

The Results section can be reduced as the diagrams are extensive and convey the important information

The English needs attention --for instance there are sentences beginning with abbreviations and the word patien should be reserved for humans (see Oxford Dictionary)

Author Response

Dear reviewer,

Thank you for constructive comments concerning our manuscript entitled “Proximal perineural femoral nerve injection in pigs using an ultrasound-guided lateral subiliac approach – A cadaveric study”. We have studied your comments carefully and made a correction which we hope to meet with your approval. We answer your questions or comments in details in the following texts. Detailed answer to review:

The Paper is far to long and needs extensive editing.

The Introduction is rambling and should be  significantly reduced in length as should the Discussion and concentrate on the main points of the findings

-Thank you! Done!

The Results section can be reduced as the diagrams are extensive and convey the important information

-We reduced Introduction and Discussion section. We would like to leave the result section as it is. Thank you!

The English needs attention --for instance there are sentences beginning with abbreviations and the word patien should be reserved for humans (see Oxford Dictionary)

Two English academic writers have checked the Text. Thank you! 

We really hope manuscript modification can meet with your approval. Thank you very much.

Sincerely,

XXX